# GhBOP1 as a Key Factor of Ribosomal Biogenesis: Development of Wrinkled Leaves in Upland Cotton

**DOI:** 10.3390/ijms23179942

**Published:** 2022-09-01

**Authors:** Yanwen Wang, Zhimao Sun, Long Wang, Lingling Chen, Lina Ma, Jiaoyan Lv, Kaikai Qiao, Shuli Fan, Qifeng Ma

**Affiliations:** 1State Key Laboratory of Cotton Biology, Institute of Cotton Research of Chinese Academy of Agricultural Sciences, Anyang 455099, China; 2College of Life Sciences, Shaanxi Normal University, Xi’an 710062, China; 3Hainan Yazhou Bay Seed Lab, Sanya 572000, China; 4National Nanfan Research Institute (Sanya), Chinese Academy of Agricultural Sciences, Sanya 572000, China

**Keywords:** wrinkled leaves, upland cotton, cell cycle, ribosomal biogenesis, *GhBOP1* gene

## Abstract

Block of proliferation 1 (BOP1) is a key protein that helps in the maturation of ribosomes and promotes the progression of the cell cycle. However, its role in the leaf morphogenesis of cotton remains unknown. Herein, we report and study the function of *GhBOP1* isolated from *Gossypium hirsutum*. The sequence alignment revealed that BOP1 protein was highly conserved among different species. The yeast two-hybrid experiments, bimolecular fluorescence complementation, and luciferase complementation techniques revealed that GhBOP1 interact with GhPES and GhWDR12. Subcellular localization experiments revealed that GhBOP1, GhPES and GhWDR12 were localized at the nucleolus. Suppression of *GhBOP1* transcripts resulted in the uneven bending of leaf margins and the presence of young wrinkled leaves by virus-induced gene silencing assay. Abnormal palisade arrangements and the presence of large upper epidermal cells were observed in the paraffin sections of the wrinkled leaves. Meanwhile, a jasmonic acid-related gene, *GhOPR3,* expression was increased. In addition, a negative effect was exerted on the cell cycle and the downregulation of the auxin-related genes was also observed. These results suggest that *GhBOP1* plays a critical role in the development of wrinkled cotton leaves, and the process is potentially modulated through phytohormone signaling.

## 1. Introduction

Ribosomes are one of the most important organelles present in eukaryotes and prokaryotes. There are composed of rRNA and ribosomal proteins [1]. As a highly complex and evolutionarily conserved organelle, they play an important role in cell division and plant development. As previously reported, ribosomes help alleviate stress, promote the development of seedlings [2], and embryo growth [3], promote cell proliferation [1], and decrease fertility [4]. Ribosome biogenesis primarily occurs in the nucleolus, and the process is a key metabolic mechanism that occurs in a proliferating cell and it can be regulated for cell growth and proliferation [5]. The activity of more than 200 proteins is regulated via a controlled mechanism associated with ribosome biogenesis [6,7].

The biogenesis of the 60S ribosomal subunit is controlled by the Block of Proliferation *1* (BOP1). BOP1 modulates the process through functional interactions with the nucleolar protein pescadillo (PES) and WD Repeat Domain 12 (WDR12) in plants. The PeBoW complex is formed in mammalian cells when PES interacts with BOP1 and WDR12. The complex mediates the pre-rRNA processing to realize the synthesis of 5.8S and 28S rRNAs. Dysfunction results in the defective assembly of the 60S large ribosomal subunit [1,8]. PeBoW is a ribosomal biogenesis factor that significantly affects the biogenesis of ribosomes and the progression of the cell cycle in various fungi and animals [9]. BOP1 is a conserved nucleolar protein in eukaryotes and a pivotal regulator of the process of cell proliferation. It was firstly isolated from mice, and truncation of the full-length protein resulted in cell cycle arrest [10]. In vivo and in vitro suppression of the proliferation process of gastric cancer (GC) cells resulted from *BOP1* silencing. The cell cycle was blocked at the G0/G1 phase, and the apoptosis of the GC cells was triggered via the upregulation of p53 and p21 [11].

A high degree of developmental defect associated with defective cell division processes is observed in plants under conditions of a 50% decrease in transcript accumulation of the *AtBOP1* gene [12]. Defects in rRNA processing were observed in RNAi *Fragaria vesca* lines. Abnormal levels of cell proliferation were observed in *Arabidopsis thaliana* [12]. A complete loss of *BOP1* may be lethal. Reduced expression levels of *Arabidopsis* T-DNA insertion mutants and *Fragaria vesca* RNAi lines result in the formation of deformed leaves. Impairment in the process of gametogenesis and other morphological defects are also formed under these conditions. Senescence of the *Fragaria vesca* RNAi lines is observed post-flowering [12]. Defective ribosome biogenesis could be attributed to transient silencing of *BOP1* in tobacco. The process resulted in a significant reduction in the level of protein synthesis, and in abnormal leaf development and delayed cell growth. The phenotypes were similar to those observed in *PES* and *WDR12*-silenced lines [9,13].

Cotton is one of the most important cash crops. Leaf morphology influences photosynthesis, transpiration, and crop yield, so it is important for the cotton leaves to have a normal morphology. Leaf shrinkage is an important indicator of leaf morphology. It is a complex quantitative trait that is controlled by multiple genes (such as *HYPONASTIC LEAVES1* (*HYL1*), *miR166*, and *REVOLUTA (REV)*) and abiotic factors [14,15]. Ribosome-associated mutants are typically characterized by stunted growth, reduced fertility, and narrow leaves. These features were observed in the *Arabidopsis rps18*, *rps13*, and *rps5* mutants [16]. The lack of the *BRX1* gene associated with the maturation of the 60S subunit in *Arabidopsis* results in the formation of pointed rosette leaves [17]. It has been reported that the APUM23 protein present in *Arabidopsis* plays pleiotropic roles during plant development and contributes to pre-rRNA processing. Synergistic interactions between the *apum23-3* mutant and other leaf polarity mutants are also observed. These interactions alter the expression levels of leaf polarity genes and influence the process of proliferation of division-competent cells [18]. In addition, the 60S associated ribosome biogenesis factor *LSG1-2* and the *NSN1* mutants are associated with severe developmental defects, including triple cotyledons and the formation of upward curled leaves. Thus, it can be inferred that the process of ribosome biogenesis is associated with early plant and leaf development [18,19].

The functions of *BOP1* in *Arabidopsis thaliana*, *Oryza sativa*, *Fragaria vesca*, and *Nicotiana benthamiana*, have been studied. It is noteworthy that the specific functions of *BOP1* in cotton, especially upland cotton, are largely unknown. We studied *GhBOP1* expression and the functions of the gene following the virus-induced gene silencing (VIGS) procedure to understand its roles in cotton. The paraffin sections were analyzed, and the results revealed that the upper epidermis cells were larger than the lower epidermis cells. It was also observed that the mesophyll cells palisade cells were long and closely arranged. *GhBOP1* silencing resulted in cell cycle disruption, upregulation of the jasmonic acid-related genes, and downregulation of the auxin-related genes. The results reported herein indicate that *GhBOP1* is involved in cotton shrinkage. The results can potentially pave the pathway for functional studies of the morphology of cotton leaves to provide theoretical support for preventing cotton yield decline caused by cotton leaf shrinkage.

## 2. Results

### 2.1. GhBOP1 Bioinformatics and Expression Analysis

Tetraploid cotton (*Gossypium hirsutum*, Zhongmiansuo 24) was used as the sample to amplify the full-length cDNA of *GhBOP1*. Results from sequence analysis revealed that the gene consisted of 18 exons and 17 introns. In total, 729 amino acids with three C-terminal WD repeats and an expected N-terminal BOP1 domain were encoded (Appendix A). BOP1 proteins are characterized by the presence of a highly conserved domain in different kingdoms (Figure 1A). This proves its importance in the existence of animal, plant, and even microbial species. From the amino acid sequence aligned among those eleven species, these sequences show significantly high homology, the functional residues are almost unchanged, and the sequences possess the same secondary structure (Appendix A). Phylogenetic clustering of BOP1 from cotton and diverse species depicts conserved motifs within this protein. The protein consisted of a BOP1 domain and several WD40 repeat units (Figure 1B,C).

The presence of *GhBOP1* transcripts was observed in the cotton tissues under study. The protein was found in seeds, flowers, ovules, fibers, etc. Analysis of the transcriptome data revealed that the abundance of the *GhBOP1* transcripts in emerging seed germination and developing pistils was higher than in other tissues (Figure 1D).

### 2.2. Interaction of GhBOP1 with GhPES and GhWDR12

Interactions between BOP1 and PES/WDR12 have been reported in other species, and whether it exists in upland cotton is unknown. We carried out yeast two-hybrid assays to investigate whether GhBOP1 interacts with GhPES and GhWDR12 in upland cotton. The results revealed that the yeast cells carrying AD-GhBOP1 and BD-GhPES, BD-GhWDR12 grew well, while the single-gene yeast cells could not grow on the medium (SD\-Leu\-Trp\-His\-Ade + X-α-Gal) (Figure 2A). The Split-LUC (Figure 2B) and BiFC (Figure 3B) experiments were conducted to further confirm the interaction between GhBOP1 and GhPES/GhWDR12. The results revealed that only the combinations of GhBOP1 and GhPES/GhWDR12 were present, which resulted in fluorescence properties. Blue clones and fluorescence were absent in other combinations. In summary, consistent results were obtained by conducting the three experiments, and it was observed that GhBOP1 could interact with GhPES and GhWDR12 in upland cotton.

### 2.3. Subcellular Localization of GhBOP1 and GhPES/GhWDR12

To further explore the spatial expression of the *GhBOP1* and *GhPES*/and *GhWDR12* genes, we conducted a subcellular localization experiment. In the experimental group, *GhBOP1* and *GhPES*/*GhWDR12* were predominantly expressed in the nucleolus of tobacco leaves. A faint signal could also be detected in the nucleoplasm (Figure 3A). Results from BiFC experiments revealed that GhBOP1 interacted with GhPES and GhWDR12 in the nucleus (Figure 3B). The results reflect the functions of GhBOP1 and GhPES/GhWDR12 and reveal that they contribute to in the process of ribosome biogenesis.

### 2.4. Phenotype of VIGS in GhBOP1 in Cotton

To study the role of *GhBOP1* in cotton, we conducted the virus-induced silencing assay with the *GhBOP1* gene and analyzed its phenotype. Compared to the blank control group, the *TRV2:GhBOP1* experimental group grew slowly. Under these conditions, the leaf wrinkled, and the leaf edge curved unevenly downward. In addition, the phenotype was found in the third true leaf (Figure 4A). During the growth of silent lines, the phenotype of the leaf folds became inconspicuous. Analysis of the paraffin sections of the leaves in the experimental group revealed that the cells in the upper epidermis of the leaves were larger than those in the lower epidermis of the leaves. The palisade tissue in the mesophyll cells did not exhibit orderly cylindrical cells in the control group. The presence of highly disordered and long cells was observed (Figure 4C,D). The real-time quantitative PCR technique was used for the analysis of leaves, and the results revealed that the silencing efficiency of *GhBOP1* was higher than 50%, and the maximum efficiency recorded was 63% (Figure 4B). This indicated that the silencing efficiency of *GhBOP1* was high. Therefore, it can be inferred that the changes in mesophyll cells in the leaves of the *GhBOP1*-silent lines could be attributed to the significantly low level of *GhBOP1* expression.

### 2.5. Expression of Jasmonic Acid, Cell Cycle, and Auxin Related Genes in GhBOP1-Silenced Lines

The real-time quantitative PCR technique was used to explore the expression of GhBOP1 interaction genes, cell cycle-related genes, jasmonic acid-related genes, and auxin-related genes in the *GhBOP1* silenced strains. It was observed that the expression levels of *GhPES* and *GhWDR12* were significantly low in the *GhBOP1*-silenced lines (Figure 5C). The levels of *GhE2Fa*, *GhE2Fb*, and *GhRBR* in the E2F-RBR pathway, and the levels of the s-phase-specific expression genes associated with the cell division cycle 6 (*GhCDC6*), *GhORC1B*, chromatin licensing, DNA replication factor 1A (*GhCDT1A*), and *GhCDT1B* were significantly reduced. The expression levels of the CycD family *GhCYCD1;1* were significantly downregulated. Though the expression level of *GhCYCD3;1* decreased, the decrease was not significant, suggesting a reduced degree of cell proliferation (Figure 5A). Furthermore, the genes associated with auxin biosynthesis (such as tryptophan aminotransferase related 2 (*GhTAR2*)) and those associated with auxin response (such as *GhGH3.3*) were significantly downregulated. The gene 12-oxo-phytodienoic acid reductase 3 (*GhOPR3*) was associated with the biosynthesis of jasmonic acid, and the expression level of this gene was significantly upregulated (Figure 5B). Collectively, these results suggest that the wrinkled cotton leaves belonging to the *GhBOP1*-silenced lines may be modulated via phytohormone signaling.

## 3. Discussion

The nucleolus is an important nuclear structure present in eukaryotic cells. The formation of ribosomes occurs initially in the nucleolus, and the process involves a series of complex reactions that require the participation of many nucleolar-related factors. Abnormal structural abnormalities in ribosomes are observed in the nucleolus. These abnormalities result in apoptosis, the arrest of the cell cycle, and cellular senescence. Severe developmental defects, such as the formation of aberrant vein patterns, the formation of pointed first leaves [20], the appearance of triple cotyledons, and the upward curling of leaves can be attributed to the blockage of ribosomal biogenesis [21]. Defective biogenesis of the 60S large ribosomal subunit could be attributed to the VIGS of the PeBoW genes present in *Nicotiana benthamiana* [9]. Downregulation of *PES*, *BOP1* or *WDR12* results in cell cycle arrest and altered expression of cell cycle-related genes, thereby affecting cell division and proliferation [22,23], and it often produces abnormally developed leaves [9]. Flow cytometry analysis in *Arabidopsis Atbop1* mutants revealed that *Atbop1* mutants do possess a greater number of polyploid cells [12]. We confirmed that GhBOP1 interacted with the GhPES and GhWDR12 proteins. Furthermore, we observed that the levels of *GhPES* and *GhWDR12* were significantly reduced in the *GhBOP1*-silenced lines (Figure 5C).

In plants, ribosomal abnormalities can result in a variety of phenotypes. Furthermore, ribosomal abnormalities can result in the tipping and narrowing of leaves, delayed root development, insensitivity to auxin, and delayed embryonic development. Photo-perception and photosynthesis primarily occur in leaves, and the functions of leaves significantly affect plant growth. In most plants, the leaf functions as a solar panel where carbon dioxide is converted to water, carbohydrates, and oxygen during photosynthesis [23,24]. Leaves contain three principal axes, and the processes of differentiation, expansion, and cell division are controlled by various intrinsic genetic programs [25]. The morphogenesis of leaves is closely modulated by hormonal and intrinsic genetic factors. Analysis of various series of events reveals that small and simple molecules that function as plant hormones significantly affect the processes associated with the development and growth of plants [26]. Cotton leaves are formed above the ground, and they occupy a large area. Cotton leaves are the primary organs of the plant. Photosynthesis occurs in the leaves resulting in the production of energy. *GhBOP1* silencing resulted in changes in the morphology and degree of flatness of cotton leaves. This can be potentially attributed to the inhibition of the establishment of leaf polarity due to impaired ribosome function [27]. A balance between polarity, auxin response, and cell division is essential for the efficient formation and development of normal and flat leaves. Any imbalance will result in a change in the leaf shape and crimped, wrinkled, twisted, curled, radiating, or shrunken leaves may be formed under such conditions [14,15,28,29]. The changes in leaf structure during the middle and late periods can be attributed to the changes in the mechanisms that coordinate the cell division and/or expansion processes of the dorsal and ventral leaves. The phenotype with curling of the leaves upward can be potentially associated with the enlargement of cells present on the dorsal surface of the leaf [28,30]. The adaxial/distal mesophyll cells in the leaf were studied. It was observed that, the abaxial epidermal cells present in the upward-curled leaf were larger than the abaxial epidermal cells. Narrow and long cells were absent in the dorsal epidermis. The palisade and sponge mesophyll cells look like oblate spheres parallel to the upper epidermis, and the palisade cells and sponge cells have slightly different cell sizes and chloroplast content [15]. Silencing of the *GhBOP1* gene resulted in wrinkling of cotton leaves and a slight downward curve of the leaf margins. Analysis of the histological sections revealed that the upper epidermal cells were larger than the lower epidermal cells, the palisade tissue was elongated and dense (Figure 4A,B). The BOP1 and its interacting gene (PES, WDR12) also exhibited a similar phenotype in tobacco. The growth of the *TRV:NbBOP1*, *TRV:NbPES* and *TRV:NbWDR12* tobacco leaves was hindered, and abnormal development of the leaves was observed. Small, wrinkled, and twisted leaves containing localized yellow patches were present [13]. The development was arrested, and premature aging in *Arabidopsis thaliana* was observed under conditions of dexamethasone-induced RNAi of *BOP1* and *WDR12*/*PES*. Kinetic studies were conducted, revealing that the depletion of the PeBoW protein resulted in defective ribosome biogenesis. This resulted in the inhibition of the processes of cell division and expansion. The processes also resulted in cell differentiation [9,13]. Ribosomal-protein-defective mutants show impaired cell proliferation in leaf mesophyll cells, and a strong cell proliferation defect results in larger leaf mesophyll cells [27,31]. The effect of virus induced *GhBOP1* gene silencing on the morphology and smoothness of cotton leaves can be potentially attributed to uneven cell division and expansion that result in changes in the growth rate of dorsal ventral and dorsal leaves.

The growth and development of plants are controlled by plant hormones, among which auxin and jasmonic acid play important roles in the processes of cell division, cell expansion, and cell differentiation [32,33,34,35]. A coordinated mechanism exists between auxin and ribosome biogenesis that regulates plant growth and development [36]. Most of the ribosome mutants were characterized by various abnormal phenotypes related to auxin. Altered expression levels of numerous genes associated with auxin biosynthesis, transport, and signal transduction were observed [32,33,34,35]. In *GhBOP1*-silenced plants, the expression levels of the auxin biosynthesis-related genes (*GhTAR2* and *GhGH3.3*) were significantly lowered, and the expression level of the jasmonic acid biosynthesis-related gene *GhOPR3* was significantly elevated (Figure 5B). The expression of the major cell cycle regulators, such as *E2Fa*, *CycD*, and *CDKA*, and the G1/S transition process are regulated by this class of hormones [37,38]. The stability of the *E2Fb* protein increased under conditions of auxin modulated proteolysis [39]. The progression of the cell cycle is inhibited by the treatment of jasmonic acid in synchronized tobacco BY-2 cells. The process is inhibited when G1/S and G2/M transitions are blocked [40]. The growth of the *Arabidopsis* leaf is suppressed under the action of methyl jasmonate. Growth suppression can be realized by inhibiting the processes of cell proliferation and expansion. Under these conditions, the life cycle of the leaf cells is arrested in the G1 phase [41]. Auxin also controls the process of leaf morphogenesis [30].

Plant development requires strict control of the cell cycle because the production and stabilization of plant tissue and organ structures require coordination between cell division and differentiation. In virus-induced *GhBOP1*-silenced plants, the transcription levels of *GhE2Fa*, *GhE2Fb*, and *GhRBR* were decreased significantly in the E2F-RBR pathway. The transcription levels of the s-phase-specific genes, including *GhCDT1A*, *GhCDT1B*, *GhCDC6*, and *GhORC1B*, and the transcription levels of the *CycD* family (*GhCYCD1;1* and *GhCYCD3;1*) were decreased significantly (Figure 5A). The E2F-RBR pathway (including *E2Fa*, *E2Fb*, and *RBR*) plays an important role in cell cycle progress by regulating the process of G1/S transformation in plants [42,43,44,45]. The retino-blastoma-related (RBR)/E2F/DP pathway is a major regulator of the balance between cell division, differentiation, and endocycle. The balance is achieved by controlling the G1/S transition process [43]. Under the action of mitogenic stimuli, cyclin/CDK complexes phosphorylate *RBR*, which dissociates from the E2F/DP complex that subsequently induces gene expression. This, in turn, induces the expression of the cell cycle-related genes [46]. It has been reported that the expression of *CYCD3*;1 is regulated by the availability of nutrients and auxin [47].

The virus-induced *GhBOP1* gene silencing blocks the auxin biosynthesis pathway. This results in decreased *CYCD3;1* and *CDK* transcription levels and decreased stability of the *E2Fb* protein (Figure 6). Thus, the transcription of the *E2F* target genes is associated with cell cycle progression, DNA replication and S-phase entry, and chromatin dynamics are inhibited [3,48,49]. An increase in the level of jasmonic acid also inhibits cell division and differentiation. These result in the low transcription levels of the genes associated with the S-phase. This, in turn, hinders the cell cycle and inhibits the processes of cell division and differentiation. This can potentially result in uneven cell division and expansion, resulting in changes in the growth rate of dorsal ventral and dorsal leaves. The smoothness and morphology of the leaves were affected, resulting in the formation of wrinkled young cotton leaves. Uneven bends were observed at leaf margins. Cotton leaves shrinking and irregular growth always influence the yield and the loss can be formidable in this case. We believe, in the future, that this might help find a way to prevent leaf shrinking and protect the harvest, not just of cotton, but of other plants that have the same mechanism.

## 4. Materials and Methods

### 4.1. Plant Materials and Growth Conditions

*Gossypium hirsutum* Linn. (cv CCRI24) plants were grown in the greenhouse of the Institute of Cotton Research of the Chinese Academy of Agricultural Sciences at 28 °C. VIGS cotton and *Nicotiana benthamiana* were grown in the greenhouse at 24 °C, and the 16-h light/8-h dark cycle was kept.

### 4.2. Bioinformatics Analysis

Sequence analysis, evolutionary tree analysis, gene structure and conserved motif analysis of GhBOP1 (*Gossypium hirsutum*, ZJU), XP_016718684.2; HsBOP1 (*Homo Sapiens*), BAA09473; AtBOP1 (*Arabidopsis thaliana*), AAD25679; OsBOP1 (*Oryza sativa*), BAC84089; ApBOP1 (*Auxenochlorella protothecoides*), XP_011395468.1; CrBOP1 (*Chlamydomonas reinhardtii*), A8ID74; CeY48B6A (*Caenorhabditis elegans*), Y48B6A.1; DmNOP1 (*Drosophila melanogaster*), NP_611270.1; XlBOP1 (*Xenopus laevis*), NP_001080358.1; Erb1p (*Saccharomyces cerevisiae*), NP_013764; MmBOP1 (*Mus musculus*), NP_038509; and NcBOP1 (*Neurospora crassa*), XP_330757.1 were carried out by methods reported by Carvalho et al. [12] and Sun et al. [50]. The sequence of the BOP1 genes was obtained from the NCBI website (www.ncbi.nlm.nih.gov).

### 4.3. Yeast Two-Hybrid Assays

The full-length cDNAs corresponding to *GhBOP1* (*GhWDR12* and *GhPES*) were cloned. The cloned cDNAs were inserted into the pGBKT7 (BD) and pGADT7 (AD) vectors, respectively. SnapGene was used for sequencing and data analysis. Subsequently, the AD and BD vectors were mixed and co-transformed into competent Y2HGold yeast cells by means of PEG/LiAc transformation. Following this, the yeast cells were incubated on SD-Trp/Leu plates. The cells were allowed to grow for 3 days in an incubator at a temperature of 30 °C. The best mono-clones were selected and inoculated onto SD-Trp/Leu/His/Ade + X-α-Gal plates. The results were recorded after 3 days. The experiments were conducted thrice to confirm the repeatability of the results.

### 4.4. Split Luciferase Complementation Assay (Split-LUC)

The full-length CDS of *GhBOP1* and *GhPES*/*GhWDR12* were fused to luciferase N-terminal fragment (nLUC) and Luciferase C-terminal fragment (cLUC), respectively. The recombinant plasmid was transformed following the sequencing process into Agrobacterium tumefaciens GV3101 (Weidi, China). The transient expression of tobacco was studied following the methods reported by Li et al. [36]. The leaves were removed, and fluorescein was applied to the back of the leaves prior to observation. The sample was placed in the dark for 10 min on a low-light-cooled charge-coupled device (CCD) imaging apparatus Lumazone_1300B (Roper Bioscience, Pittsfield, MA, USA) with its back facing up. The exposure time was set to 10 min. The sample was observed and photographed, and the image was captured.

### 4.5. Bimolecular Fluorescence Complementation (BiFC)

*GhWDR12* and *GhPES*, the full-length CDS of *GhBOP1*, were connected to the pSPYCE-35S and pSPYNE-35S carriers, respectively. After sequencing, the vector was transferred into *Agrobacterium tumefaciens* GV3101, following which the transient expression of tobacco was studied. The sample was cultured for 48 h, and the fluorescence was observed using the aDmi8 inverted microscope (Leica, Wetzlar, Germany).

### 4.6. Subcellular Localization of GhBOP1 and GhWDR12/GhPES

The full-length CDS of *GhBOP1* and *GhPES*/*GhWDR12*, were inserted into green fluorescent protein (GFP) vectors, respectively. The samples were sequenced, following which the recombinant plasmid was transferred into Agrobacterium tumefaciens GV3101. Subsequently, it was mixed with H2B-RFP (positive control of nuclear localization) and injected into tobacco leaves. The inoculated tobacco plants were cultured following the 12 h dark/24 h light cycle. GFP fluorescence was observed using the aDmi8 inverted microscope.

### 4.7. Virus-Induced Gene Silencing of BOP1 in Cotton and Paraffin Section of Leaf

A sequence of *GhBOP1* was inserted into the TRV2 (pYL156) vector. The relevant sample was sequenced before transforming it into the *Agrobacterium* strain LBA4404 (Weidi, China). Shaking tables were used to culture *TRV2:CLA1* (positive control), *TRV2:00* (negative control), pYL192 (subsidiary vector), and *TRV2:GhBOP1* (VIGS). The sample was centrifuged and subsequently resuspended when the OD600 of the bacterium liquid was = 1.8. The pYL192 strain was mixed with the sample in a ratio of 1:1 (OD600 = 1.8). Following this, the sample was injected into *CCRI24* cotyledons. The *Agrobacterium*-injected plants were placed in a light-tight incubation box for 18 h. The plants were cultured in a greenhouse with a light–dark cycle (light: 16 h, 25 °C dark: 8 h, 22 °C).

Two weeks after the injection of the *Agrobacterium* strain, 1 cm-long leaves of the *TRV2:GhBOP1* and *TRV2:00* plants were cut off with scissors. The samples were fixed using the FAA fixative solution for approximately 24 h. The leaves were sent to the Biopple Company (China) to prepare paraffin sections. Image-pro Plus 6.0 (Media cybernetics, Inc., Rockville, MD, USA) was used to determine the size of the cells in the paraffin sections. The measurements were taken five times, and each time the length of a different section of the sample was measured. Statistical analysis was conducted, and column charts presenting cell size data were generated using GraphPad Prism 8 (GraphPad Software, San Diego, CA, USA).

### 4.8. Real-Time Quantitative PCR Detecting

RNAprep Pure Plant kit (Vazyme, Nanjing, China) was used for extracting RNAs from cotton leaves. StarScript II First-Strand cDNA Synthesis Mix With gDNA Remover (Genstar, Beijing, China) was used for the reverse-synthesis of cDNA from the RNA samples. Real-time quantitative PCR detection was conducted using MonAmp SYBR Green qPCR Mix (Monad, Hangzhou, China) with gene-specific primers (Appendix A) and the ABI 7500 real time PCR system. The internal controls were selected to be *GhUBQ7* and *GhHistone3*, which were the stably expressed genes in upland cotton. Three technical replicates and three independent biological replicates were recorded. The 2^−ΔΔCt^ method was used for analyzing the experimentally obtained data [51]. The mean SD values were calculated by analyzing three independent biological replicates.

## 5. Conclusions

The nucleolar protein BOP1 is conserved in plants, animals, and yeast. We report that GhBOP1 is primarily located in the nucleoli and interacts with GhPES and GhWDR12, which are also located in the nucleoli. The silencing of the *GhBOP1* gene resulted in the wrinkling of cotton leaves and the uneven bending of leaf margins. Paraffin sections were analyzed, and the results revealed that the cells in the upper epidermis were larger than those in the lower epidermis. The mesophyll and palisade cells were long and closely arranged. *GhBOP1* silencing resulted in cell cycle disruption and downregulation of cell cycle-related genes. Upregulation of the genes associated with jasmonic acid and the downregulation of the genes associated with auxin were also observed. These results suggest that *GhBOP1* can potentially influence the growth and development of cotton leaves by participating in the cell cycle process. This is similar to the role of *BOP1* in other plants, animals, and yeast. The results can potentially pave the pathway for functional studies of the morphology of cotton leaves, while providing theoretical support for preventing cotton yield decline caused by cotton leaf shrinkage.

## Figures and Tables

**Figure 1 ijms-23-09942-f001:**
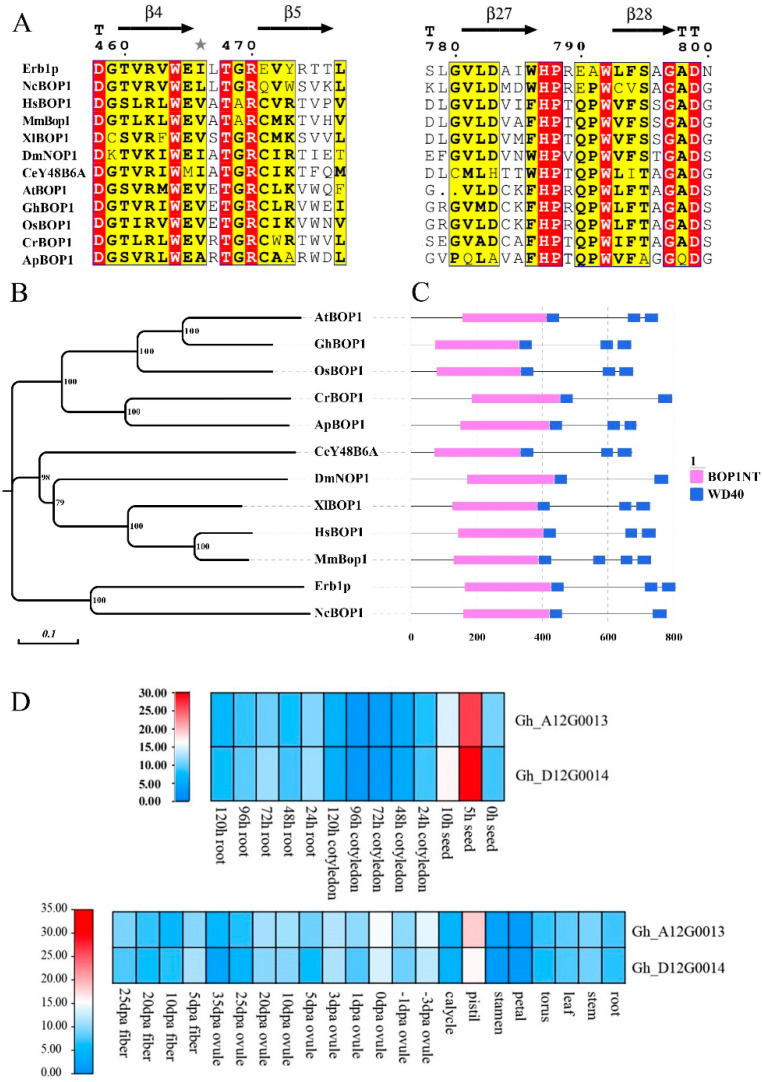
Characterization of Block of Proliferation 1 (BOP1) in different species and expression of *GhBOP1* in cotton at different periods. (**A**) The highly conserved amino acid residues of BOP1 in different species are highlighted (red box). (**B**) Adjacency tree of BOP1 between different species, based on 1000 repeated lead values displayed next to branch points. (**C**) Domain structure composition of BOP1 proteins in different species: the two conserved motifs in BOP1 protein are represented by pink (BOP1NT) and blue (WD40 repeat) boxes, respectively. (**D**) Expression of *GhBOP1* in different time periods and tissues. Gh_A12G0013 is the *GhBOP1* gene number on chromosome A12, and Gh_D12G0014 is the *GhBOP1* gene number on chromosome D12.

**Figure 2 ijms-23-09942-f002:**
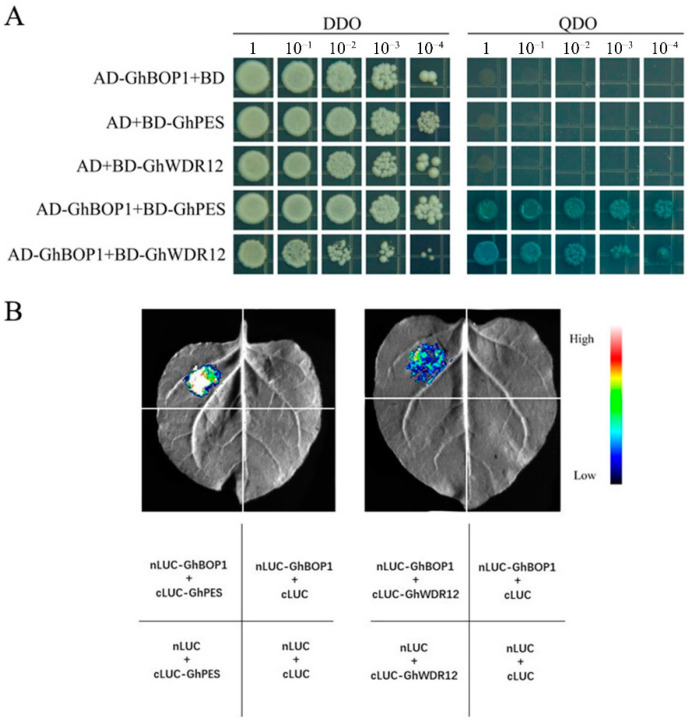
Interaction of GhBOP1 with GhPES and GhWDR12. (**A**) The interaction of GhBOP1 with GhPES and GhWDR12 was verified by yeast two-hybrid method, and blue yeast indicated that they interacted. DDO represents the SD-Trp/Leu medium, and QDO represents the SD-Trp/Leu/His/Ade + X-α-Gal medium. (**B**) Interaction of GhBOP1 with GhPES and GhWDR12 was verified by conducting the luciferase complementation experiment, and their interaction was indicated by fluorescence.

**Figure 3 ijms-23-09942-f003:**
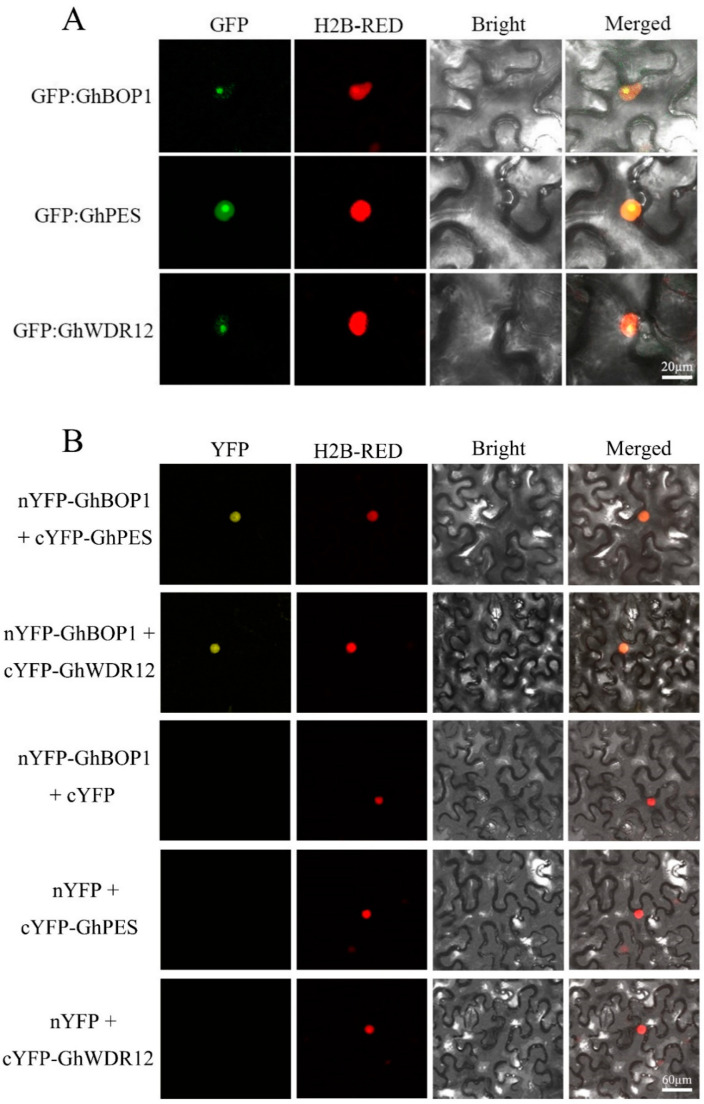
Spatial expression of the *GhBOP1* and *GhPES*/*GhWDR12* genes. (**A**) Green fluorescence refers to the location of *GhBOP1* and *GhPES*/*GhWDR12* in tobacco (located in the nucleolus), red fluorescence to the positive control of nuclear localization, and yellow fluorescence refer to the overlapping region of the green fluorescence and red fluorescence (located in the nucleolus), respectively. (**B**) GhBOP1 interacts with GhPES and GhWDR12. Yellow fluorescence refers to the interaction of GhBOP1 with GhPES and GhWDR12. Red fluorescence (positive control of nuclear localization) and orange fluorescence refer to the overlapping region of yellow and red fluorescence, respectively.

**Figure 4 ijms-23-09942-f004:**
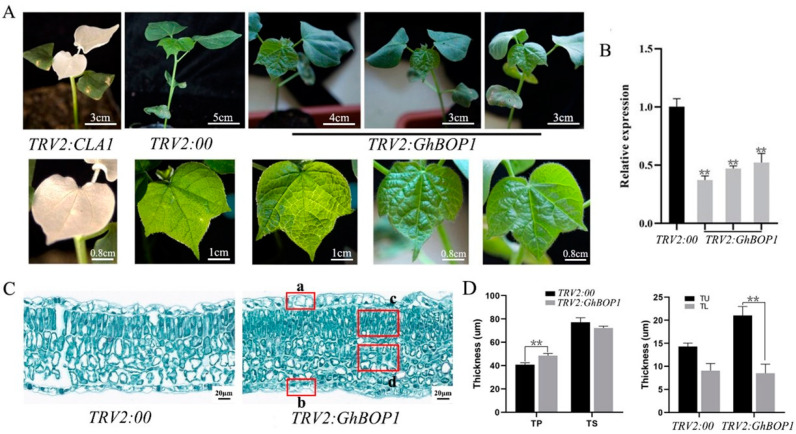
Silencing of *GhBOP1* changed the flatness and morphology of cotton leaves. (**A**) Phenotypes of blank control and gene silenced plants. Plants from left to right are positive control (*TRV2:CLA*), negative control (*TRV2:00*) and three silent *GhBOP1* lines (*TRV2:GhBOP1*). (**B**) *GhBOP1* expression level in blank control and VIGS plants. *GhHistone3* was used as an internal reference gene for data normalization. (**C**) Paraffin-sectioned phenotypes of leaves of blank control plants and wrinkled leaves of *GhBOP1*-silenced plants. a presents the upper epidermal cell, b presents the lower epidermal cell, c presents the palisade tissue, and d presents the spongy tissue (red box). (**D**) Statistics of cell thickness corresponding to the paraffin section of leaves. TU represents the thickness of the upper epidermal cells, TL represents the thickness of the lower epidermal cells, TP represents the thickness of the palisade tissue, and TP represents the thickness of spongy tissue. Asterisks denote statistical significance as follows: ** *p* ≤ 0.01.

**Figure 5 ijms-23-09942-f005:**
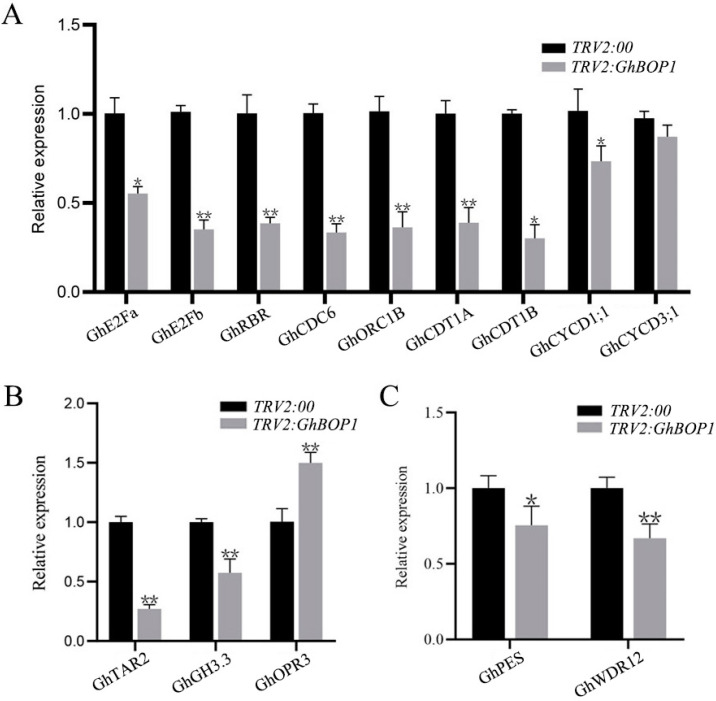
Expression of interacting genes, cell cycle-, auxin- and jasmonic acid-related genes in *GhBOP1-*silenced lines during early response. (**A**) Expression of cell cycle-related genes. (**B**) Expression of auxin- and jasmonic acid-related genes. (**C**) Expression of *GhBOP1* interacting genes. The data were tested by conducting the one-way analysis of variance (ANOVA) with the aid of the Student’s *t*-test at the significant level as follows: * *p* ≤ 0.05; ** *p* ≤ 0.01. The error bars indicate standard error from three independent repeated trials.

**Figure 6 ijms-23-09942-f006:**
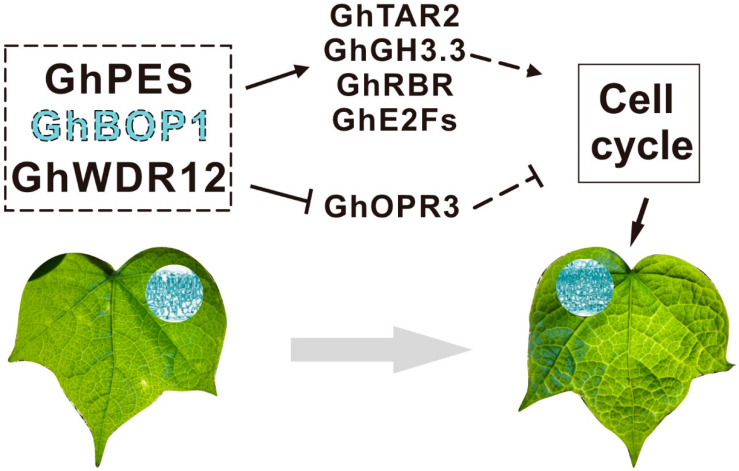
Pathway diagram of *GhBOP1* associated with cell cycle progression. The dotted box represents the PeBoW protein complex, and the dotted line *GhBOP1* represents gene silencing.

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
