# Peer review of "GhBOP1 as a Key Factor of Ribosomal Biogenesis: Development of Wrinkled Leaves in Upland Cotton"

_ijms, 2022, doi:10.3390/ijms23179942_

Round 1
Reviewer 1 Report
Wang and his colleagues present role of GhBOP1 involved in development of Wrinkled Leaves in Upland Cotton. The aims of this study are relatively well achieved by GhBOP1 proteins identification and Functional characterization. The conclusions are supported by the data, and the manuscript is well organized. However, I have several concerns that should be addressed before publication.
Page7, Figure 4B. statistical significance (0.01 or 0.05 level) should be declared in the note of figure. In addition, the clear of Figure 4C should be improved.
Page7, Figure 5. The “Û”represent the significant level of P < 0.05? "ÛÛ”represent the significant level of P < 0.01? the author should explain in the note of Figure.
Moreover, the English should be improved in the whole manuscript before publication. I recommend revising English grammar by any of proofreading companies or native speaker. Several corrections are given below, but these are only examples.
Page 2, L53. "first" should be changed to "firstly".
Page 2, L87. Make sure that the Latin name names of species are in italic format and the species names are in Roman type. On Line 87, “rice” and “ Strawberry” should be in Roman type, there are only the example, please check the whole manuscript. In addition, italic text of “Arabidopsis” should be unified in the whole manuscript.
Page3, L107. " have "suggest to change to " show " .
Page5, L126 . “BOP1 and PES, and WDR12” suggested to change to “BOP1 and PES / WDR12”. In addition., the similar sentence in Page5, from L133 to L134. Please check the whole manuscript and correct them in the revised manuscript.
Page7, L162 . "TRV2:GhBOP1” should be in italic format. Please check the whole manuscript.
Reviewer 2 Report
The authors' Yanwen Wang, Zhimao Sun, Long Wang, Lingling Chen, Lina Ma, Jiaoyan Lv, Kaikai Qiao, Shuli Fan, and Qifeng Ma article « GhBOP1 as A Key Factor of Ribosomal Biogenesis: Develop- 2 ment of Wrinkled Leaves in Upland Cotton», is devoted to one of the most important economic crops- cotton investigation, in this regard, the relevance is difficult to dispute.
Disadvantages.
The introduction necessary to correct.
Avoid repetitions in the text. Make the text more logical and consistent. Highlight the purpose of the study more clearly and better reveal the further practical application of the obtained results.
The results are described satisfactorily.
Correct the discussion and make it more consistent and logical, to emphasize what exactly the importance of the obtained results.
Conclusion. Focus on practical application obtained results.
Author Response
1. The introduction necessary to correct. Avoid repetitions in the text. Make
the text more logical and consistent. Highlight the purpose of the study more
clearly and better reveal the further practical application of the obtained
results.
Reply: Thank you very much for your careful reading and valuable suggestion. We modified some sentences to make the text more logical and consistent, and to highlight the purpose of the study more clearly and better reveal the further practical application of the obtained results in the revised manuscript, as follows:
Page2, line 40: delete the sentence “the ribosome performs translational functions in cells.”;
Page2, line 44: delete the phrase “The process of ribosome biogenesis”;
Page2, line 44: the sentence “The process of ribosome biogenesis is a key metabolic mechanism that occurs in a proliferating cell and it can be regulated for cell growth and proliferation. Ribosome biogenesis primarily occurs in the nucleolus” modified to “Ribosome biogenesis primarily occurs in the nucleolus, and the process is a key metabolic mechanism that occurs in a proliferating cell and it can be regulated for cell growth and proliferation”;
Page2, line 75: delete the sentence “Changes in the nucleoli structures were observed,”;
Page4, line 78-80: add sentences “Cotton is one of the most important cash crops.” and “So it's important for the cotton leaves to have a normal morphology.”;
Page5, line 108: add sentence “To provide theoretical support for preventing cotton yield decline caused by cotton leaf shrinkage.”
2. Correct the discussion and make it more consistent and logical, to
emphasize what exactly the importance of the obtained results.
Reply: Thank you very much for your careful reading and valuable suggestion. We modified some sentences to make it more consistent and logical, to emphasize what exactly the importance of the obtained results in the revised manuscript.
Page17, Line 316 “The growth and development of plants are controlled by plant hormones, among which auxin and jasmonic acid play important roles in the processes of cell division, cell expansion, and cell differentiation” was moved ahead of “A coordinated mechanism exists between auxin and ribosome biogenesis that regulates plant growth and development” in line 310;
Line 318: delete the sentence “Auxin modulates plant growth and development, and controls cell expansion and division.”;
Line 330: delete the sentence “As a highly complex and evolutionarily conserved
organelle, ribosomes perform translation functions in cells and significantly influence the processes of cell division, growth, and development.”;
Line 360: add sentences “Cotton leaves shrinking and irregular growth always influence the yield and the loss can be formidable in this case. We believe, in the future, might help find a way to prevent leaf shrinking and protect the harvest of not just cotton, but also other plants that have the same mechanism.”
The results are described satisfactorily.
3. Conclusion. Focus on practical application obtained results.
Reply: Thank you very much for your careful reading and valuable suggestion. We modified some sentences to focus on practical application obtained results in the revised manuscript. Such as, page24, Line471-473, add sentence, it reads “The results can potentially pave the pathway for functional studies of the morphology of cotton leaves, while providing theoretical support for preventing cotton yield decline caused by cotton leaf shrinkage.”
Reviewer 3 Report
Dear editor and colleagues,
I have read with great interest the submitted manuscript “GhBOP1 as a key factor of ribosomal biogenesis: Development of wrinkled leaves in upland cotton” submitted in the ijms.
It is a study that focuses on the interaction of Block of proliferation 1 (BOP1) protein with GhPES and GhWDR12 proteins and the phenotypical alterations in upland cotton leaves.
The authors have generally used adequate and appropriate techniques (two-hybrid systems, Split Luciferase Complementation Assay, qPCR, as well as, provided evidence of Subcellular localization of GhBOP1, GhWDR12 and GhPES in tobacco, while have produced gene silencing (BOP1) lines in cotton).
The manuscript is rather well focused, clear and a few typos or nomenclature errors exist (for instance L87 taxa are in italics; Arabidopsis, rice, Strawberry etc)
As a result, the manuscript has merit for publication, since it is (largely) scientifically and grammatically solid
I have however two main concerns that the authors should address (preferably with extra data)
The authors focused on leaf structure while these proteins can affect the plant in whole. Did you detect any deficiencies in roots or shoots? Did you find any differences in gene transcription of the involved proteins? I find it difficult to believe that gene silencing affects only leaves’ morphology (unless GhBOP1 has a tissue specific pattern)
Also, the authors claim that alterations can be attributed to changes in cell cycle; but unfortunately, do not provide data (besides transcription) on cell cycle or endopolyploidy that is often associated to leaf morphology. I suggest using flow cytometry and study cell cycle/endopolyploidy on these cotton genotypes because transcription of genes does not ‘tell the whole story’
Based on the above reasons I recommend a major revision
Reviewer 4 Report
Upland cotton (Gossypium hirsutum L.) represents over 90% of world cotton production. Cotton textiles were known in ancient India and Peru and since the invention of cotton gin, this material became widespread and extremely popular. Current world production of cotton is about 25 million t and it occupies 2.5% of world arable land.
Leaves shrinking and irregular growth always influence the yield and the loss can be formidable in this case. The paper "GhBOP1 as A Key Factor of Ribosomal Biogenesis: Development of Wrinkled Leaves in Upland Cotton" by Yanwen Wang, Zhimao Sun, Long Wang, Lingling Chen, Lina Ma, Jiaoyan Lv, Kaikai Qiao, Shuli Fan and Qifeng Ma studies the block of proliferation 1 (BOP1) protein, known to affect leaf shrinking in other plants, but not investigated in cotton yet, using gene silencing and PCR to study selected metabolites and their interactions.
Obtained results support authors' hypothesis that BOP1 plays and important role in growth and development of cotton leaves; they also found that up-regulation of the genes associated with jasmonic acid and down-regulation is associated with auxin.
The paper is given in good, simple scientific English, methods and findings are clearly described, hypothesis is based on previous reports about other species, that are properly cited, and the results are in good accordance with the expectations. It is an important topic that, in the future, might help find a way to prevent leaf shrinking and protect the harvest of not just cotton, but also other plants that have the same mechanism.
Mostly, the paper is neat and could stay as it is. Please fix the text "(Error! Reference source not found)" at the line 395 and check the text carefully for small details (i.e. line 381, too much space before 8h).
After this minor revision, the paper is recommended to be accepted in IJMS; it will be both interesting and beneficial to the readers.
Author Response
Thank you so much for your careful check.. We have checked the text carefully for small details and corrected them in the revised manuscript. And, in Page13, Line 395 “(Error! Reference source not found.)” have been changed to “(Table S1)”.
Round 2
Reviewer 3 Report
The authors have provided reasonable justifications and the manuscript has been improved. I recommend an acceptance of the ms as it stands